# Structure and proposed DNA delivery mechanism of a marine roseophage

Yang Huang[1,2,7], Hui Sun[1,2,7], Shuzhen Wei[3,7], Lanlan Cai[4], Liqin Liu[1,2], Yanan Jiang[1,2], Jiabao Xin[1,2], Zhenqin Chen[1,2], Yuqiong Que[1,2], Zhibo Kong[1,2], Tingting Li [1,2], Hai Yu[1,2], Jun Zhang [1,2], Ying Gu [1,2], Qingbing Zheng [1,2] ✉, Shaowei Li [1,2] ✉, Rui Zhang [3,5] ✉ & Ningshao Xia [1,2,6] ✉

Tailed bacteriophages (order, *Caudovirales*) account for the majority of all phages. However, the long flexible tail of siphophages hinders comprehensive investigation of the mechanism of viral gene delivery. Here, we report the atomic capsid and in-situ structures of the tail machine of the marine siphophage, vB_DshS-R4C (R4C), which infects *Roseobacter*. The R4C virion, comprising 12 distinct structural protein components, has a unique five-fold vertex of the icosahedral capsid that allows genome delivery. The specific position and interaction pattern of the tail tube proteins determine the atypical long rigid tail of R4C, and further provide negative charge distribution within the tail tube. A ratchet mechanism assists in DNA transmission, which is initiated by an absorption device that structurally resembles the phage-like particle, RcGTA. Overall, these results provide in-depth knowledge into the intact structure and underlining DNA delivery mechanism for the ecologically important siphophages.

Bacteriophages, the most abundant biological forms in the biosphere, contribute significantly to shaping microbial diversity, mediating genetic exchange, and modulating cycling of biogeochemical elements[1–4]. All known phages possess a proteinaceous capsid for genome encapsulation, and most deploy a specialized tail apparatus for recognizing and opening an entry point on the cell envelope for genome translocation[5–8]. Tailed phages (*Caudovirales*) can be classified into three distinct types: myophage, with a contractile tail (e.g., T4, Mu); podophage, with a short tail (e.g., P22, T7); and siphophage, with a long non-contractile tail (e.g., SPP1, λ)[9].

Most tailed phages have an icosahedral capsid mainly formed by multiple copies of the major capsid proteins (MCPs). The first structure was solved for the siphophage, HK97, and this identified a new protein fold referred to as the HK97-fold, which has subsequently been found in the capsid proteins of various other phages and herpesviruses, despite a low sequence identity[10]. On the capsid of the tailed phages, one of the fivefold vertices is replaced by a portal complex that connects to the phage tail. Podophages sequentially assemble a stubby tail on the completed head, whereas myophages and siphophages have distinct phage assembly pathways, with their more sophisticated tails formed by multiple tail components interacting in a strict order, connecting to the phage capsid via head-to-tail connector[11,12]. Siphophages and myophages share a common tail-binding machinery, with two basic head completion proteins that are distinguished as adaptors or stoppers based on whether they extend or reversibly close the portal[12,13]. For tail assembly, an absorption apparatus initially forms,

[1]State Key Laboratory of Molecular Vaccinology and Molecular Diagnostics, School of Public Health, School of Life Sciences, Xiamen University, Xiamen 361102, China. [2]National Institute of Diagnostics and Vaccine Development in Infectious Diseases, Xiamen University, Xiamen 361102, China. [3]State Key Laboratory of Marine Environmental Science, Fujian Key Laboratory of Marine Carbon Sequestration, College of Ocean and Earth Sciences, Xiamen University, Xiamen 361102, China. [4]Department of Ocean Science, The Hong Kong University of Science and Technology, Hong Kong, China. [5]Institute for Advanced Study, Shenzhen University, Shenzhen 518060, China. [6]Research Unit of Frontier Technology of Structural Vaccinology, Chinese Academy of Medical Sciences, Xiamen 361102, China. [7]These authors contributed equally: Yang Huang, Hui Sun, Shuzhen Wei. ✉e-mail: abing0811@xmu.edu.cn; shaowei@xmu.edu.cn; ruizhang@xmu.edu.cn; nsxia@xmu.edu.cn

which is dedicated to specific recognition and irreversible interactions with host receptors (e.g. lipopolysaccharide, teichoic acid, and porins)[14–16] and differs considerably in complexity from simple tail fibres to intricate constructions of tail spikes or baseplates[7,17–19]. The apparatus then primes the polymerization of the cylindrical tail, which comprises three essential components: a central core built of stacked rings of the tail tube proteins (TTPs) capped by the terminator proteins, inside which the tape measure proteins (TMPs) are positioned[11,20–24]. Although rough ideas of the mechanisms of DNA delivery and virus assembly of phages are elucidated, given the structural and genetic diversity of phage tails, the precise processes associated with phage assembly, infection, and DNA delivery remain elusive for most phages.

The *Roseobacter* clade is a dominant group of marine bacteria widely distributed in coastal and open waters, surface and deep oceans and sediments, and is important in the global biogeochemical climate[25–28]. Accordingly, phages infecting the *Roseobacter* clade (roseophages) are widespread in the ocean and are considered as major biotic factors influencing the biology, ecology, and biogeochemistry of the *Roseobacter* clade[29–31]. So far, more than 50 roseophages have been isolated and sequenced but none has been structurally characterized[32]. We recently characterized a novel roseophage, vB_DshS-R4C (R4C)[33], that could infect the *Dinoroseobacter shibae* DFL12[T], a ubiquitous group of Gram-negative α-proteobacteria of the *Roseobacter* clade[34]. R4C phage is a distinct member of the siphophage family, as determined via phylogenetic and comparative genomic analyses[33].

In this study, we use cryo-electron microscopy (cryo-EM) and structure prediction to determine the atomic structure of the phage R4C capsid and the in-situ structure of the tail machinery to characterize the distinguished long rigid tail. The capsid and the tail tube protein are resolved at near-atomic resolutions of 3.63 Å and 3.43 Å, respectively. The C12 reconstruction of the portal-adaptor complex, with an overall resolution of 4.7 Å, characterizes an adaptor architecture that decorates an accessory Ig-like domain at its periphery. Specific positional patterning of the negative charge distribution within the tail tube provides evidence for the previously identified ratchet mechanism that assists in DNA transmission[35]. Moreover, we identify an absorption device with structural features not previously reported for other siphophages but resembles that of the phage-like particle of the gene transfer agent of *Rhodobacter capsulatus* (RcGTA)[36]; but the former equips an extra peripheric domain in the megatron protein. Overall, our work solves and explores in depth the intact structure and DNA delivery mechanism of an ecologically important siphophage.

## Results and discussion
### Overall capsid structure
Raw cryo-EM images of the R4C phage show a virus head with a diameter of ~66 nm and a long non-contractile tail with a length of ~110 nm (Supplementary Fig. 1A). Using these data, we sought to determine the cryo-EM structure of the R4C capsid head, resolved to an overall resolution of 3.63 Å by cryo-EM single-particle analysis (Fig. 1A, Supplementary Fig. 1 and Supplementary Table 1). The R4C capsid shell exhibits a $T = 7$ quasi-icosahedral lattice with a largest diameter of ~664 Å (Supplementary Fig. 1B). The capsid comprises 415 copies of the MCP (vBDshSR4C_006), organized into 11 penton vertexes and 60 hexons. The atomic models of the MCPs within one asymmetric unit containing one hexon plus one pentameric monomer were then built (Fig. 1B, C). Through structural superimposition, we find that the overall structure of the R4C MCP is highly similar to HK97 gp5 despite a low sequence identity (Supplementary Fig. 2, Fig. 1D). Similar to the canonical HK97-fold, the R4C MCP is composed of a long N-terminal arm (N-arm), an axis domain (A-domain), a periphery domain (P-domain), and an extended loop (E-loop). Structural differences

between MCPs of R4C and HK97 are mainly seen in the A-domain, with loops of residues 155–167 and 202–218 of the R4C MCP deflected inward compared with that of HK97 (Fig. 1D). Superimposition of six copies of the hexameric MCP monomer shows conformational heterogeneity, with structural variations primarily located on the E-loop and the N-arm (Supplementary Fig. 3), which contributes to accommodate the deferent radius at deferent location of the capsid shell. Such structural variations of the E-loop and N-arm are more noticeable between the hexon and penton MCPs; that is, the pentameric MCP subunit has a more bending overall in its architecture with the E-loop turning up for ~16° vs the N-arm for ~14° (Fig. 1E). This structural deviation results in the adaptive formation of hexons and pentons reflected in diameter and height. The penton presents with a protruding and narrower surface (height ~44 Å and diameter ~130 Å) compared with the hexon (height ~27 Å and diameter ~148 Å) (Fig. 1F). Both penton and hexon are assembled by similar inter-capsomeric interaction patterns mediated predominantly by two adjacent A-domains. Of note, the long E-loop spans the outer surface of the MCP toward the neighbouring P-domain, and the tip of the E-loop adjoins the distal N-arm of another spaced monomer, forming a head-to-head circulating interaction mode (Fig. 1F); such interactions are common in the capsid of phages[37–39].

### Interaction pattern between the capsomers
To further explore the structural stability of R4C capsid assembly, we focused on the detailed inter-capsomeric interactions. A hierarchical network of penton-hexon and hexon-hexon interactions mediate the assembly of the R4C capsid (Fig. 2A). The penton-hexon interactions are mediated primarily by the two anti-parallel N-arms from neighbouring capsomers (Fig. 2B); these types of interactions are also observed in hexon-hexon interactions (Fig. 2C). Besides, the hexon-hexon interactions near the icosahedral threefold and quasi-threefold axes are divisible into two layers, in which nine capsomers from three different hexons are implicated: The inner layer interactions involve P-domains of three capsomers positioned closest to the threefold axis (Fig. 2D), whereas the outer layer is held by interactions from P-domain protrusions, circulating E-loops from another three hexomeric MCPs, and three N-arms from other MCPs (Fig. 2D). Both side faces of one E-loop in the outer layer make contact with the protruding G-loop and P-loop from the inner P-domain, whereas the tip of the E-loop—together with the neighbouring N-arm—connects closely beneath the P-domain, forming a network of hydrogen bonds and salt bridges to stabilize capsomeric interactions (Fig. 2E). Of note, the capsid shell is further stabilized by strong electrostatic interactions between the two layers (Fig. 2F, G). Positive electrostatic potentials are observed on the three inner layer P-domains and at the tips of the N-arms (Fig. 2F), with negative charges noted at the contact regions of E-loops from the outer layer (Fig. 2G). Similar surface charge distribution at the same location is found in RcGTA[36] but not HK97[37] or TW1[40] (Supplementary Fig. 4). In addition, complementary electrostatic potential surfaces are also evident at the interfaces of adjacent MCP subunits within the hexon or penton (Fig. 2H), often regarded as an interaction mode for capsid assembly[41,42]. Finally, the three P-loops closest to the threefold axis make contact with each other and form a channel with a radius of 3.2 Å, which is similar to that in HK97 (2.9 Å) and RcGTA (4.0 Å), but smaller than that in TW1 (6.0 Å) (Supplementary Fig. 4). Such inter-E-loop interactions may be weak or abolished in the TW1 phage, which may explain why it requires decoration by additional proteins to stabilize the capsid. Overall, multiple interaction modes are advantageous in stabilizing the capsid of R4C.

The typical HK97-fold of MCPs is also common among capsid proteins of marine and non-marine phages. However, structure-based phylogenetic tree analysis shows that MCPs of phages from marine environments do not form a separate cluster(s) (Supplementary Fig. 5), despite their contrasting environment (e.g., salinity,

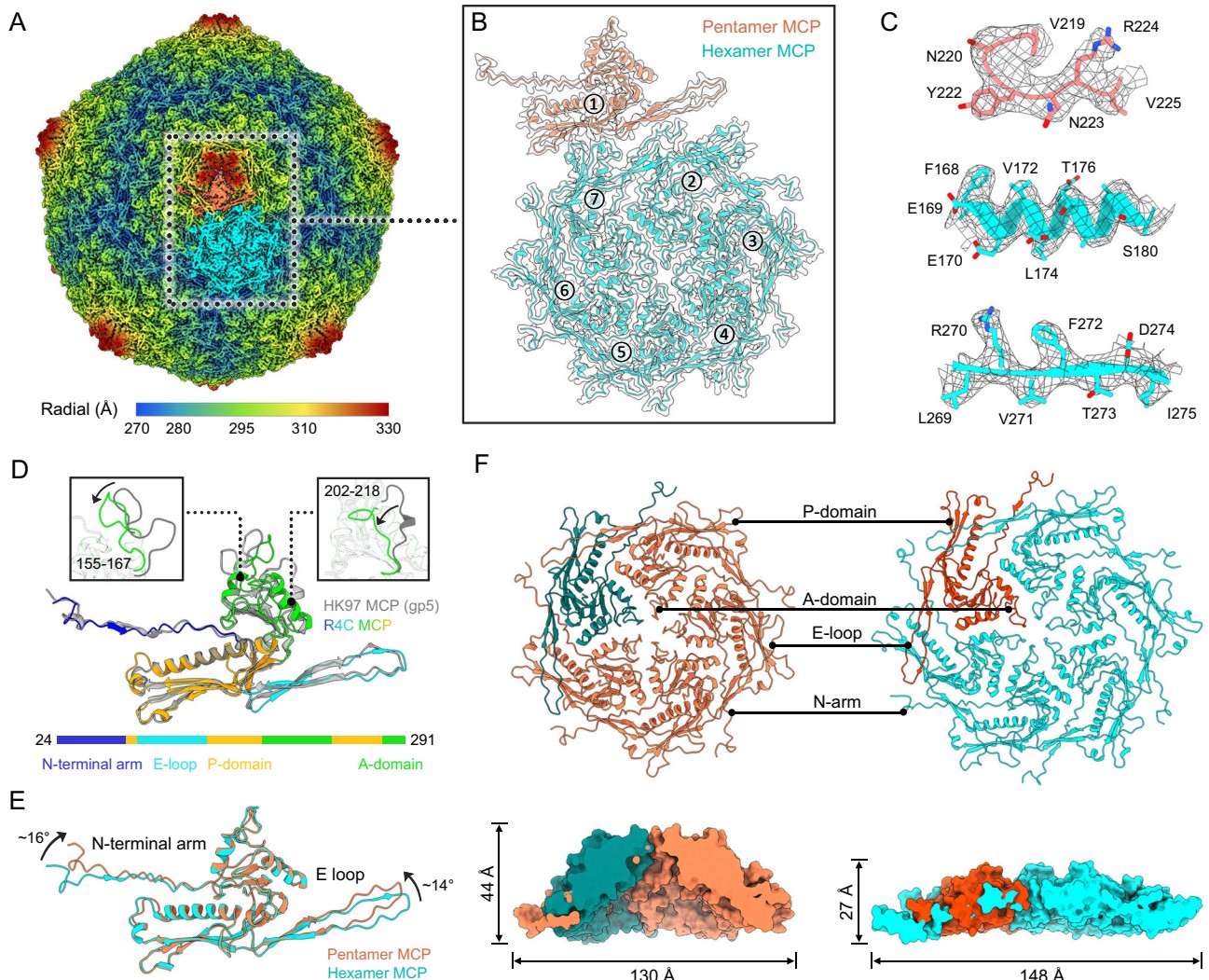

**Fig. 1 | Cryo-EM structure of the R4C capsid. A** Icosahedral reconstruction of the R4C capsid, coloured radially from the centre to the outer regions of the density map. **B** Density map of an asymmetric unit of the capsid (transparent grey) fitted with the corresponding model of the MCPs containing a pentameric monomer (orange) and a hexon (cyan). **C** Representative section density maps (grey mesh) and corresponding atomic models (sticks) of MCP illustrate the side chain features. **D** Superimposition of the R4C MCP and HK97 MCP (PDB ID: 1OHG, depicted in grey) shows their structural conservation and variation. Close-up views of the structural variations are shown on the top left and right panels. **E** Superimposition of pentamer and hexamer MCPs shows the conformational heterogeneity in the N-terminal arms and E-loops. **F** Cartoon presentations (upper panels) of the top views of a penton (left) and a hexon (right) and the corresponding clipped surface presentations (lower panels) of the side views shows their diversity in height and diameter of the protruding surface.

osmotic pressure, etc.). Thus, marine phages share a common ancestor with non-marine phages. Unlike the covalently cross-linked chain present in the capsid of HK97[37], or the formation of an additional (decoration) protein to stabilize the capsid shells found in phage BPP-1[41], TW1[40] and Mic[38], the R4C capsid is assembled by non-covalent, inter-capsomeric interactions without any additional decoration (Supplementary Fig. 4). The R4C capsid genome is regularly packaged into evenly spaced dense layers with a moderate inter-layer distance of ~24 Å (Supplementary Fig. 1B). Compared with other caudoviruses with similar $T = 7$ capsid heads, such as bacteriophage T7, lambda, TW1, P47-26[37,39,40,43–45], R4C has a normal-sized capsid (diameter, 600–660 Å) but accommodates a relatively smaller genome (36 kb, versus 40, 48 and 83 kb for T7, Lambda and P47-26, respectively). This smaller genome has a lower calculated packaging density (0.44 bp/nm$^{3}$[33,46]) than the $T = 7$ caudoviruses typical packaging density (0.54 bp/nm$^{3}$)[47]. Thus, regular and simple interaction modes (i.e., without covalent cross-link chains or decorating proteins) may act in concert with and be a consequence of the relatively smaller genome size and lower internal pressure in the R4C capsid.

## Overall structure of the R4C long tail

The R4C phage deploys a long, straight tail apparatus, which could not be reconstructed during the initial icosahedral reconstruction of the viral capsid. To gain further insight into the organization and function of these non-MCP components, we used a series of reconstruction procedures to obtain an intact structure of the R4C virion (176-nm-long) with an in-situ structure of the tail machine (119-nm-long) (Fig. 3A, B). The unique portal vertex, out of 12 fivefold vertexes on the capsid, was classified by focused classification imposing fivefold (C5) symmetry (Supplementary Fig. 6A). Further reconstruction of the portal vertex with C5 symmetry and without symmetry (C1) produced medium-resolution maps of the C5 portal vertex (6.6 Å) and the C1 portal vertex (11.5 Å) (Supplementary Fig. 6A). Two other structures were defined following several rounds of sequential localized classification and sub-particle reconstruction with different masks: the C12 dodecameric portal and adaptor (4.7 Å), and the C6 stopper and terminator (6.6 Å) (Fig. 3B, Supplementary Fig. 6A, B). Filament tracing and two-dimensional (2D) classification of the R4C tail identified a non-contractile tail, consisting of multiple layers of TTP discs around the

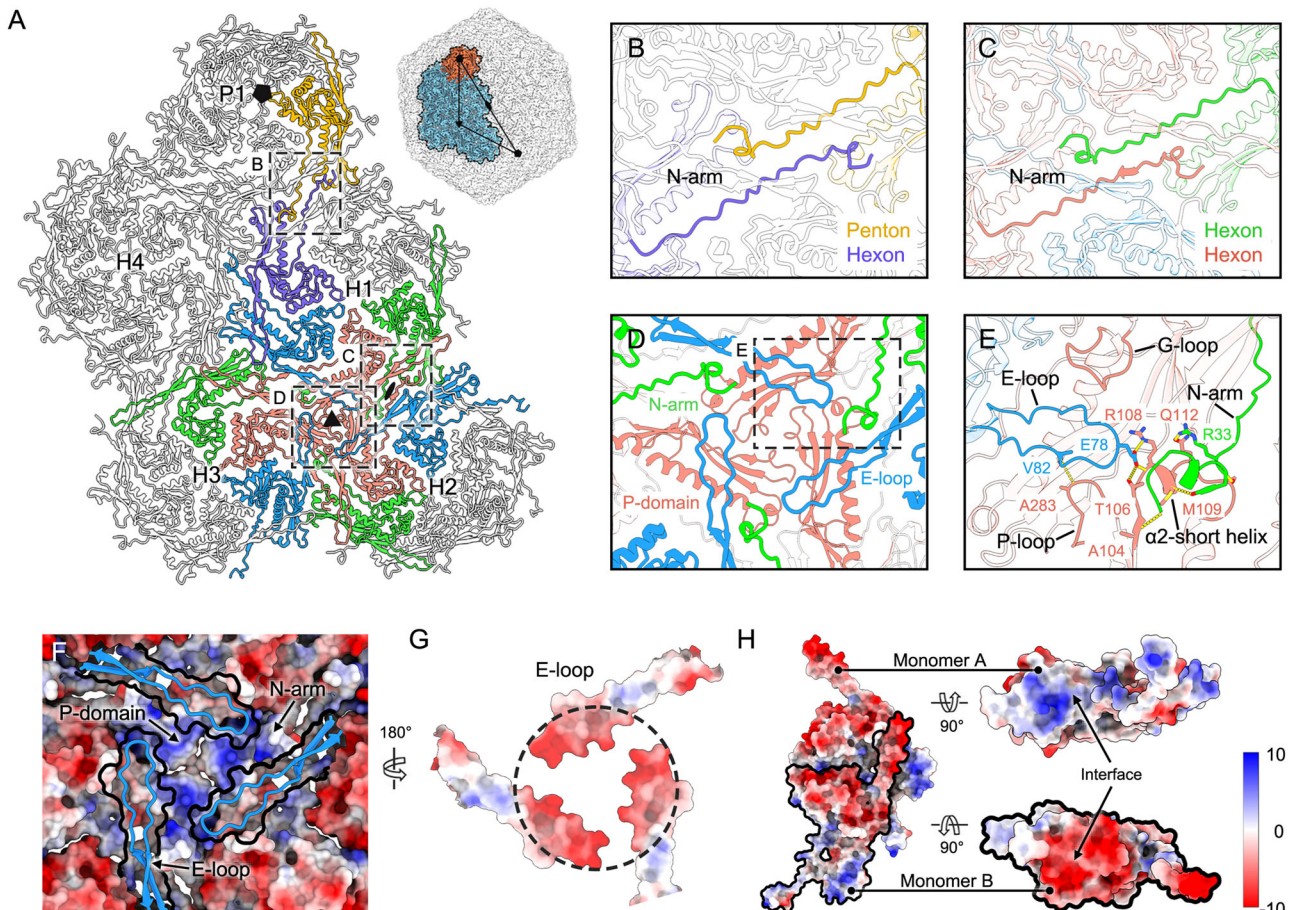

**Fig. 2 | Interaction details between capsomers. A** Capsomer network containing the interfaces around the five-, three- and twofold axes, as denoted by a black pentagon, triangle, and oval, respectively. One penton (P1) and four hexons (H1 to H4) are included in the interaction analysis. Anti-parallel N-arms are involved in the inter-capsomeric interactions between the penton and hexon MCPs (**B**) and between two hexon MCPs (**C**) via extension to neighbouring capsomers. **D**, **E** Interaction patterns near the threefold axis are viewed from outside. Interaction details for one-third of the interface in (**D**) are shown in (**E**). An interaction network of hydrogen bonds is demarcated by yellow dashed lines. **F** Electrostatic potential surface representation of the same region as in (**D**) except for the E-loops (cartoon representation). The outline of the E-loops depicted by the surface are highlighted as black lines. **G** Electrostatic potential surfaces of E-loops that observed by rotation 180° through (**F**). Complementary electrostatic potentials are observed near the tips of the E-loops. **H** Electrostatic potential surface representation of two adjacent hexamer MCP monomers. Complementary electrostatic potentials occur at the interface of monomer A (top right) and monomer B (bottom right). The electrostatic potential scale is shown in the colour bar (red–white–blue).

TMPs (Fig. 3C). Through helical reconstruction imposing C6 symmetry, we obtained a near-atomic resolution density of the tail tube, which revealed a tubular hexamer for one disc (Supplementary Fig. 6C). The distal ends of the R4C tails were re-extracted and resolved by imposing C3 symmetry, resulting in a 4.5-Å structure containing a C6 distal tail (Dit), a C3 baseplate module, and three C3 tail fibre trimers (Fig. 3B, Supplementary Fig. 6D). We then performed de novo structure predictions for each tail component using the trRosetta server (This was due to their low sequence similarity with the reported structures)[48]. All predicted models were built carefully according to the density maps and the final models fitted well into the corresponding maps, allowing for confident side chain placement of the tail tube, and unambiguous backbone placement of the other proteins (Fig. 3D).

Collectively, we report that the R4C phage is composed of 12 distinct proteinaceous components (596 proteins in total), including an unmodeled peptidase (vBDshSR4C_015) near the distal end of TMPs (Fig. 3B, E, F, Supplementary Table 2). We see that 415 major capsid proteins (vBDshSR4C_006) assemble into an icosahedral capsid, with one of fivefold vertices formed by 12 portal proteins (vBDshSR4C_005) incorporating into 12 adaptor proteins (vBDshSR4C_004). The portal extends into the capsid and is

surrounded by genomic DNA, above which is stacked the bulk of a disordered, condensed DNA along the C12 axis (Fig. 3B). The adaptor docks to the stopper hexameric ring, with the latter formed by stopper proteins (vBDshSR4C_009) that maintain DNA inside the capsid (Fig. 3B). The 88-nm-long tail is composed of an apical hexameric ring of terminator proteins (vBDshSR4C_018); a middle tail tube with a total of 120 TTPs (vBDshSR4C_010) forming 20 layers of hexamers; and a bottom hexameric ring comprising six Dit proteins (vBDshSR4C_013) (Fig. 3B, E). Below the Dit is the baseplate, consisting of 3 hub proteins (vBDshSR4C_014), 3 megatron proteins (vBDshSR4C_016) and 9 fibre proteins (vBDshSR4C_017) (Fig. 3B, E). The baseplate is presented as a trimeric hub-megatron complex connected to three claw-like trimers of fibre proteins (Fig. 3A).

**Portal-adaptor-stopper complex sticks the capsid shell and anchors genomic DNA**

As with most siphophages, the head-to-tail connector of R4C is composed of the portal, adaptor and stopper proteins organized into successive rings (Fig. 4A). The connector avoids leakage of the highly compacted DNA and provides the interaction interface for binding of the phage tail (Fig. 4A)[11].

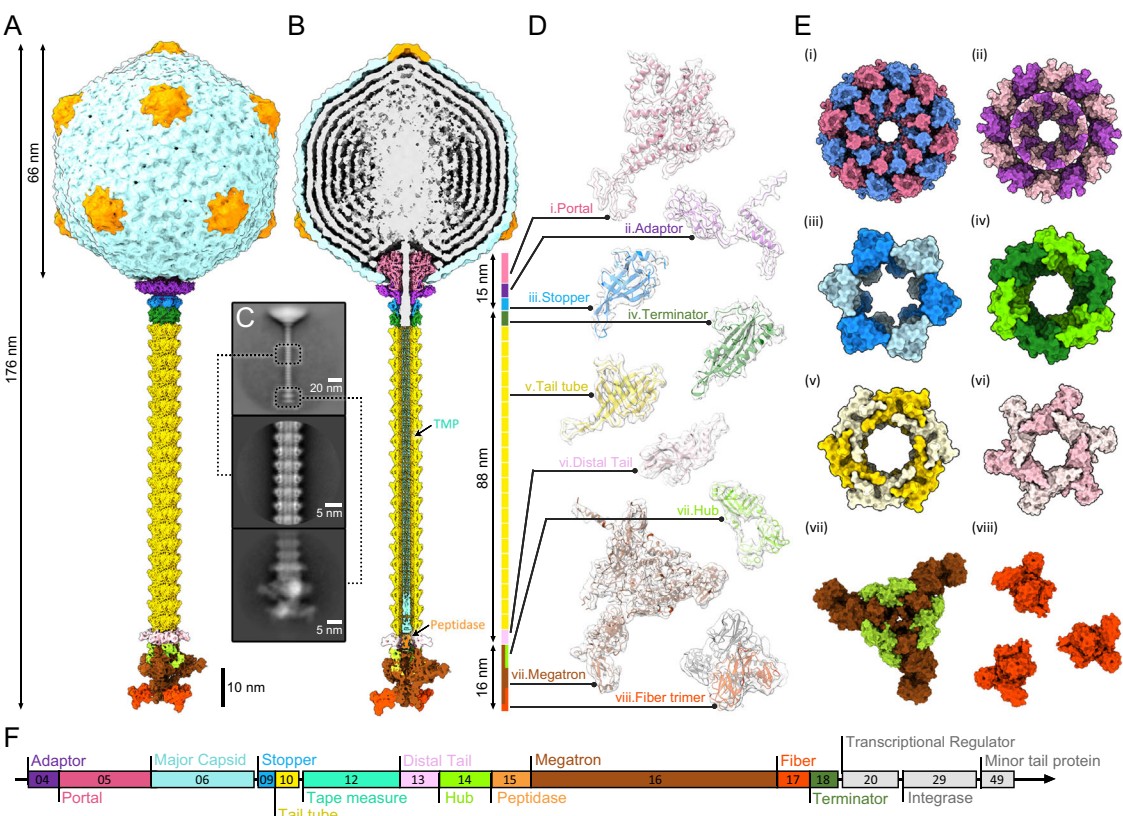

**Fig. 3 | Overall architecture of the phage R4C with in-situ structure of the tail apparatus. A** Composite density maps of the capsid and the tail apparatus are coloured with reference to proteinaceous components according to the gene map in (**F**). **B** A cut-away view of the density maps of the whole virion revealing inner structure information of the genome (grey), the portal vertex elements (pink), the tape measure proteins (TMPs, turquoise) and the peptidase (orange). Scale bar corresponds to 10 nm. **C** The 2D classification images of the whole tail (top), tail tube (middle), and baseplate (bottom). **D** For each subunit (monomer), the segmented cryo-EM density map is shown in semi-transparent grey with the fitted atomic model (ribbon). **E** Top views of the portal (i), adaptor (ii), stopper (iii), terminator (iv), tail tube (v), distal tail (vi), hub-megatron complex (vii) and tail fibres (viii) of the R4C phage. For the first six polymers, the odd-numbered and even-numbered subunits are depicted in different colours. **F** Schematic representation and assignment of the gene module encoding for the 12 structural proteins of R4C. ORFs that code for non-structural proteins are coloured in grey.

The portal complex is involved in head assembly, tail attachment, and genome packaging and releasing[49,50]. The R4C portal protein, forming a canonical cone-shaped dodecamer with a height of 10 nm, shows considerable structural similarity with portal proteins of other phages, despite poor sequence conservation (Fig. 4B, Supplementary Fig. 7). The portal protein, characterized from top to bottom, comprises separate crown (residues 476−551), wing (1−300, 394−475), stem (301−323, 377−393), and clip domains (324−376)[51] (Fig. 4A). The C-terminal crown domain consists of three α-helices (α9, α10 and α11) connected by short turns and an additional disordered 29-residue C-terminal (Fig. 4A). The α9 helices that outline the medial surface contour at the tip of the portal are radially slanted outward and present as a funnel-shaped opening toward the inner capsid; this may facilitate the release of DNA (Fig. 4B). Concomitantly, the positively and negatively charged residues lining the α9 opening, including K478, K486, K489, E485 and D479, may regulate DNA translocation (Fig. 4B). The wing domain forms the central part of the portal inside the capsid, with a long, kinked α-helix (α8) traversing the entire portal wing domain serving as a spine (Fig. 4A). A typical tunnel loop connecting α8 forms the narrowest channel with a diameter of ~30 Å (Fig. 4A, B); an analogous structural feature (e.g., phages SPP1 and T4) confines the DNA and prevents its reflux from the capsid[52,53], with the specific conformational rearrangement in the tunnel loop regulating genome release[54]. The stem domain mainly comprises two α-helices (α5 and α7) that connect the wing and clip domains (Fig. 4A). The clip domain, straddling the thick capsid wall, is partially exposed outside the R4C

head and interacts with the adaptor proteins (Fig. 4A, C). Residue K365 of the clip domain induces a positively charged ring on the internal surface of the channel outlet (Fig. 4B); in phage T4, positively charged residues at this position were thought to attract DNA during the early genome packaging period[53]. The 31 residues at the N-terminus of the R4C portal protein are disordered but presumed to be situated around the outer surface of the portal, as suggested in phage T4, and these residues may initiate further head assembly[53].

The C1 density map of the portal vertex presents with strong genomic densities of terminal DNA along the portal axis, as well as circular DNA and several layers of genomic DNA (Fig. 4C). Three regions on the portal wing domain closely interact with and anchor the genomic DNA through positively charged residues: the RRRLR motif (residues 257-274) anchors the circular DNA, like that observed in the herpesvirus capsid and such circular DNA was termed as anchor DNA[55]; R100 and K105 at the 10-residue loop (residues 99−108) and R453, K458, and K461 at the 12-residue loop (residues 452−463) of the wing domain extend and interact with the layered genomic DNA (Fig. 4D). As for the portal-capsid interaction, the α/β sub-fold at the periphery of the wing domain rests close to the capsid inner wall with a spacious interface, mediating a 5-fold to 12-fold symmetry mismatch between the portal and the capsid (Fig. 4E, Supplementary Fig. 8A, B).

The portal complex is capped by a dodecameric adaptor, which is located outside the capsid shell and interacts with both the portal and the capsid MCPs (Fig. 4C, F). There is currently no reported protein

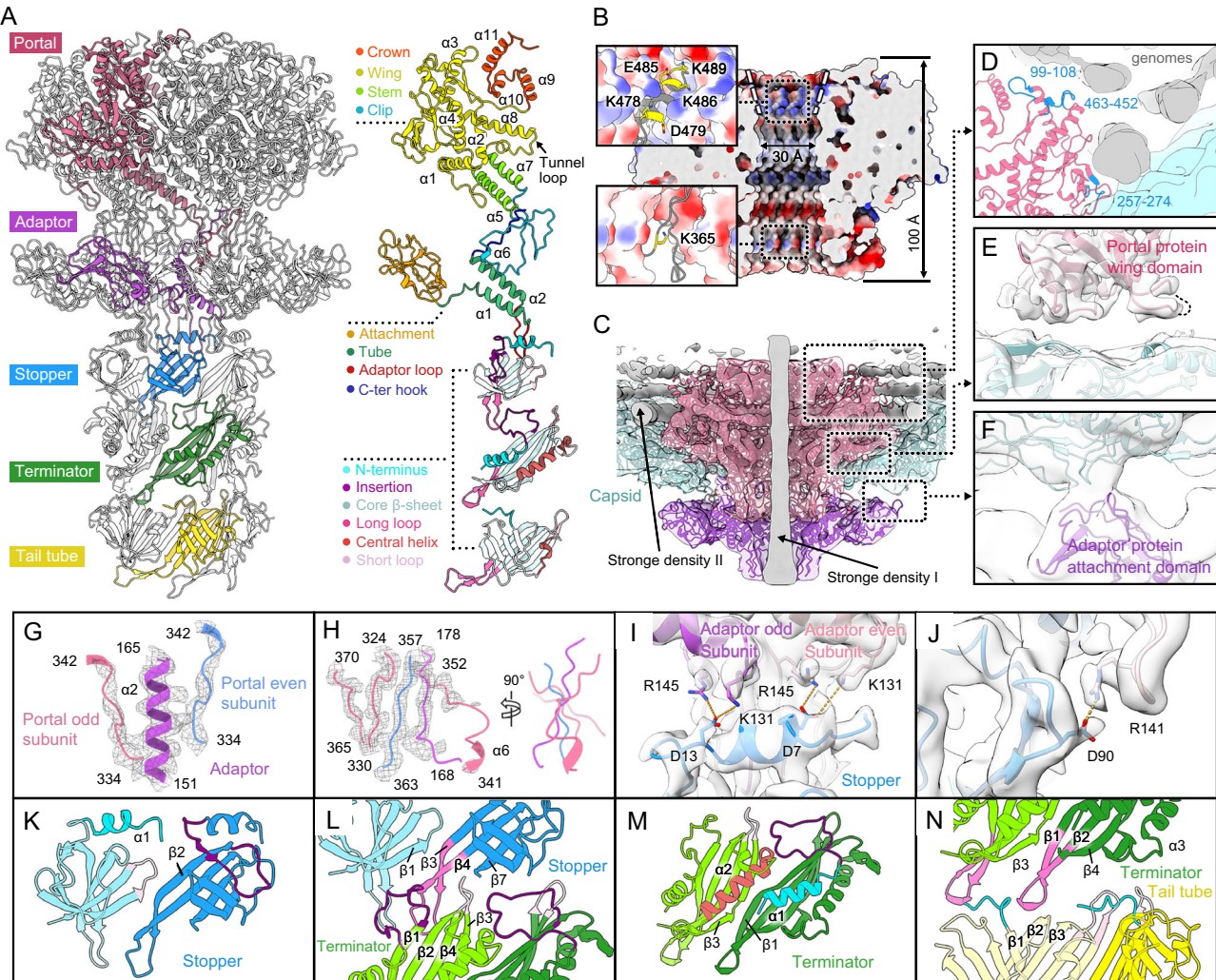

**Fig. 4 | Structure and inter-layer interactions of the head-to-tail connector.**
**A** Side view (left panel) of the head-to-tail interface composed of the portal, adaptor, stopper, terminator and a layer of tail tube. The models of each subunit are shown in ribbon and coloured with reference to the domain (right panel). An arrow indicates the tunnel loop of the portal protein. **B** Cut-away views of the electrostatic surface of the portal (positive, blue; negative, red). The two insets show the charged residues distributed on the portal interior surface formed by the crown (top) and the clip (bottom) domains. **C** Cut-away views of the composite maps of the portal vertex, comprising the capsid and the genome maps that contributed to the C1 reconstruction, and fitted with the MCPs (cyan), the portal and adaptor maps contributed from C12 reconstruction and fitted with the portal (pink) and adaptor (purple) proteins. Genome density is shown non-transparently with the other maps transparent. **D** Three loops of the portal protein involved in interactions with the genomes are highlighted in blue. Density map from the C1 reconstruction reveals the interfaces between the capsid and the portal (**E**), and the capsid and the adaptor (**F**); these were mediated by the wing domain of portal protein (**E**) and the attachment domain of adaptor protein (**F**), respectively. **G, H** Detailed interface between the adaptor (purple) and two portal proteins (distinguished by pink and blue). **I, J** Fitting of models of the stopper protein (blue) and two adaptor proteins (distinguished by purple and pink) into the C6 map (transparent surface) reveals symmetry mismatch. Residues involved in the electrostatic interactions are shown as sticks. **K–N** Intra- and inter-layer interactions in the stopper (differentiated by light and deep blue), terminator (light and deep green) and tail tube (light and deep yellow). Inter-subunit interaction of the stopper proteins and the terminator proteins are shown in (**K**) and (**M**), with inter-layer interfaces between the stopper and terminator proteins, the terminator and tail tube proteins shown in (**L**) and (**N**), respectively. Structural elements involved in intra- and inter-layer interactions are highlighted in different colours.

sharing a similar structure, as revealed by DALI search[56]. Following the nomenclature established for the RcGTA adaptor, the R4C adaptor protein can be divided as: attachment domain (1–106), tube domain (107–130, 149–166), adaptor loop (131–148) and C-terminal hook (167–178) (Fig. 4A). The N-terminus of the adaptor protein is an immunoglobulin (Ig)-like domain (attachment domain), and this domain is connected to the tube domain by a long, flexible loop (Fig. 4A). The dodecameric attachment domains are identified in a running clockwise fashion and show a symmetry-mismatched interaction within the R4C capsid around the fivefold symmetry (Fig. 4F, Supplementary Fig. 8A, C). The Ig-like domain is found in many different structural proteins of tailed phages and may be a product of rampant inter-phage horizontal gene transfer[57,58]. The phage Ig-like

domains are typically classified into three distinct Pfam (the protein families database) families—Big2, I-set and FN3—with proposed functions in facilitating phage adsorption and/or increasing infectivity[59,60]. Yet, the Ig-like attachment domain of the R4C adaptor belongs to none of these families. The proximity of the R4C adaptor to the phage head suggests that the attachment domain is unlikely to be involved in the adsorption process during infection but, instead, relates to local conformational stability. The tube domain consists of two α-helices, with the α2 sandwiched between loops from adjacent portal proteins (Fig. 4G). The tube domain extends the C-terminal hook to the portal clip domain, forming an interaction network comprising five hybrid β-sheets (Fig. 4H). The adaptor loop provides an interface for binding of the stopper proteins.

The stopper protein, a crucial module for head assembly, shares common structural features (anti-parallel β-strands barrel) with other siphophages, myophages and phage-like entities (Supplementary Fig. 9A, B)[61,62]. Notably, two diverse groups of stopper proteins can be specified based on the absence (e.g. RcGTA g7[36]) or presence (e.g. λ gpFII[61]) of a long N-terminal extension that interacts with the adaptor proteins (Supplementary Fig. 9A)[63]. The R4C stopper protein, with its long N-terminal extension, folds part of the N-terminus as an α-helix and plays a critical role in connecting the adaptor (Fig. 4A and Supplementary Fig. 9A). Specifically, the N-terminal α-helix of one stopper monomer forms a close contact with loops from two neighbouring adaptor (odd and even) monomers (Fig. 4I). Among them, residues R145 and K131 of two neighbouring adaptors, hovering above the stopper N-terminus α-helix, form a network of salt bridges with residues D7 and D13 of the stopper protein (Fig. 4I). In addition, another electrostatic interaction forms between residue D90 of the stopper core β-strand and residue R141 of the adaptor even subunit (Fig. 4J). Several hydrophobic amino acids, including three aromatic amino acids within the stopper N-terminus helices, engage in hydrophobic stacking interactions with adjacent adaptor loops (Supplementary Fig. 8D, E). Eventually, a dodecamer plug of adaptor proteins is inserted tightly into a hexamer socket of stopper proteins, resulting in symmetry mismatch between the two complexes (Supplementary Fig. 8A, F).

Intriguingly, the R4C inter-stopper contact area (~274 Å²) formed by the N-termini, short loop, insertion loop and β2, seems too small for complex formation (Fig. 4K)[64]: we speculate that, similar to other tailed phages, the stopper proteins may exist as monomers in solution and form the stopper hexamer when incorporated into the head[61,65]. Resembling the gpFII of phage λ, the R4C stopper N-terminal tail may be flexible as a monomer, and fold into helices only when correctly interacting with the adaptor proteins (Supplementary Fig. 9A)[61]. Indeed, such conformational change ensures the orderly assembly of the complete phage head[66–68]. After head assembly, the R4C stopper engages its long loop to fit inside a groove formed by the terminator, joining the head to the preassembled tail (Fig. 4L).

## Structural basis for the long and straight R4C tail tube

Previously, tail assembly in siphophages and myophages was thought to progress through an initiator complex functioning as an absorption device, followed by polymerization of the TTPs around the TMPs[12], and mounting of a hexamer cap of the terminator onto the preassembled tail at a pre-determined tail length (defined by TMPs) to prevent aberrant polymerization of the tail tube[11,69,70]. As we described, the R4C terminator operates as a docking platform for the preassembled head and tail (Fig. 4A). Despite low sequence similarity, the R4C terminator protein shares a common structural topology with that of other long-tailed phages (e.g., λ phage gpU[11], SPP1 gp17[71]), and integrates elements of a multi-stranded β-sheet barrel and two main helices at the periphery (Supplementary Figs. 9C, D and 10A). The central α-helices and β3 cling to the neighbouring terminator surface, which is composed of an N-terminal helix, an insertion domain, and a segment of β1 (Fig. 4M). The interaction between the R4C terminator and the tail tube relies chiefly on the tips of the terminator long β-hairpins binding to the minor groove, which is formed by adjacent tail tube subunits (Fig. 4N). Residue M50 in the β-hairpin tip inserts deeply into the hydrophobic cavity at the interaction interface to further stabilize layer coupling (Supplementary Fig. 10B).

Siphoviruses typically possess long, non-contractile, flexible tails that are difficult to structurally characterize. In contrast, the intrinsically rigid tail of R4C, which presents as a long, straight tubular structure of 20 layers of TTPs (Fig. 3A, B), is conducive for structural elucidation. A density map of the tail tube at 3.43 Å resolution was obtained using helical reconstruction. The map showed a spiral architecture with an axial rise of ~40 Å and a twist of 30° (Fig. 5A). We

then solved the atomic model of TTP (Figs. 5B, C), finding conservation in the main bodies of TTPs among R4C, SPP1, T4 and RcGTA, with a similar twisted β-sandwich extending a long β-hairpin (Fig. 5C, Supplementary Fig. 9E, F). These findings point to remarkable conservation of TTPs within siphophages, myophages and other phage-like particles[36,72,73].

The helical tail tube has an outer diameter of ~85 Å and a wide inner corridor of ~43 Å that displays a strong negative electrostatic potential (Fig. 5D), as observed in other TTPs, and likely facilitates DNA traffic into the host[74]. The aspartate (e.g., D27 and D108) and glutamate (e.g., E32 and E57) residues lining the tunnel form a negatively charged, almost parallel helical track through the tail tube (Supplementary Fig. 11A), equidistant spacing of ~16 Å (Supplementary Fig. 11B). Given that this value is close to the mean spacing of DNA minor and major grooves, we speculate that DNA becomes confined between the tracks and can therefore migrate quickly along the right-handed helical trajectory due to the electrostatic repulsion during DNA translocation. An analogous ratcheting mechanism has also been proposed for phage YSD1, which ratchets DNA through a helical twist inherent in the polynucleotide[35]. This peculiar molecular feature may potentiate the migratory velocity of the genome, since the initial driving force for genome release—comes from internal turgor pressure—was calculated could only elucidate the entry of ~15% of the genome length[41–43].

The inter-monomer interactions within the same TTP disc are dominated by backbone hydrogen bonds between the β-sheet layers, with two electrostatic networks for assisted binding (Fig. 5E–H). Specifically, β5 of one TTP interacts with β2 from the adjacent subunit, forming an anti-parallel β-sheet and an extensive hydrogen-bonding network (Fig. 5F). This results in an integrated 24-stranded β-barrel structure for one tail tube disc. Negatively charged residues (i.e., D67, E103 and E70) engage with positively charged residues from the adjacent subunit (i.e., R91 and K126) establishing a dense network of salt bridges (Fig. 5G). Another electrostatic interaction is formed by R118 from one subunit contacting with E43 and E32 from the long loop of the adjacent subunit (Fig. 5H). Interactions between layers involve extensive hydrophobic interactions, with the long loop of TTP in the upper layer interacting with three TTPs from the lower layers of disc (Fig. 5E). Residue T48 and two long-loop hydrophobic residues, F46 and V33, contact the hydrophobic surface formed by the first two subunits (Fig. 5I). The M38 side chain, resembling the M50 of the terminator protein (Supplementary Fig. 10B), inserts into the hydrophobic pocket formed by the former two subunits (Fig. 5J), and the bulky aromatic side chain of W36 of the long loop is sandwiched by V49 and K107 from the upper and lower discs, respectively, to further stabilize inter-layer interactions (Fig. 5K).

The R4C TMPs, showing only blurred densities in the helical reconstruction of the tail tube, locate in the lumen of the tail conduit and may exist as a trimer[74,75]. Secondary structure predictions suggest that R4C TMP comprises a helix-sheet-helix motif resembling that found in RcGTA (Supplementary Fig. 12A)[36]. The predicted TMP N-terminus has a long hydrophobic α-helix of 517 residues and, upon formation of a trimer, forms a triple-stranded core coiled-coil (Supplementary Fig. 12A, B). The C-terminal domain of R4C TMP that follows is presumed to share a comparative topology with the tail needle of phages P22 and HK60, which comprise the triple β-helix and the inverted short α-helices (Supplementary Fig. 12A, C)[76,77]. R4C TMP plays a functionally critical role during DNA translocation. Two continuous α-helical transmembrane segments (residues 277–303, 309–327) found in the middle of the R4C TMP (Supplementary Fig. 12A and 12D) may form a channel that spans the bacterial envelop to prevent genomic degradation by periplasmic endonucleases[78]. Similar TMP assistive activities are noted in other phages (e.g., TP901-1[79] and T4[6]). The C-terminal integrated apparatus is thought to be involved in membrane penetration and structural stabilization[76]. Besides, the peptidase, which causes peptidoglycan degradation[80], is inferred to

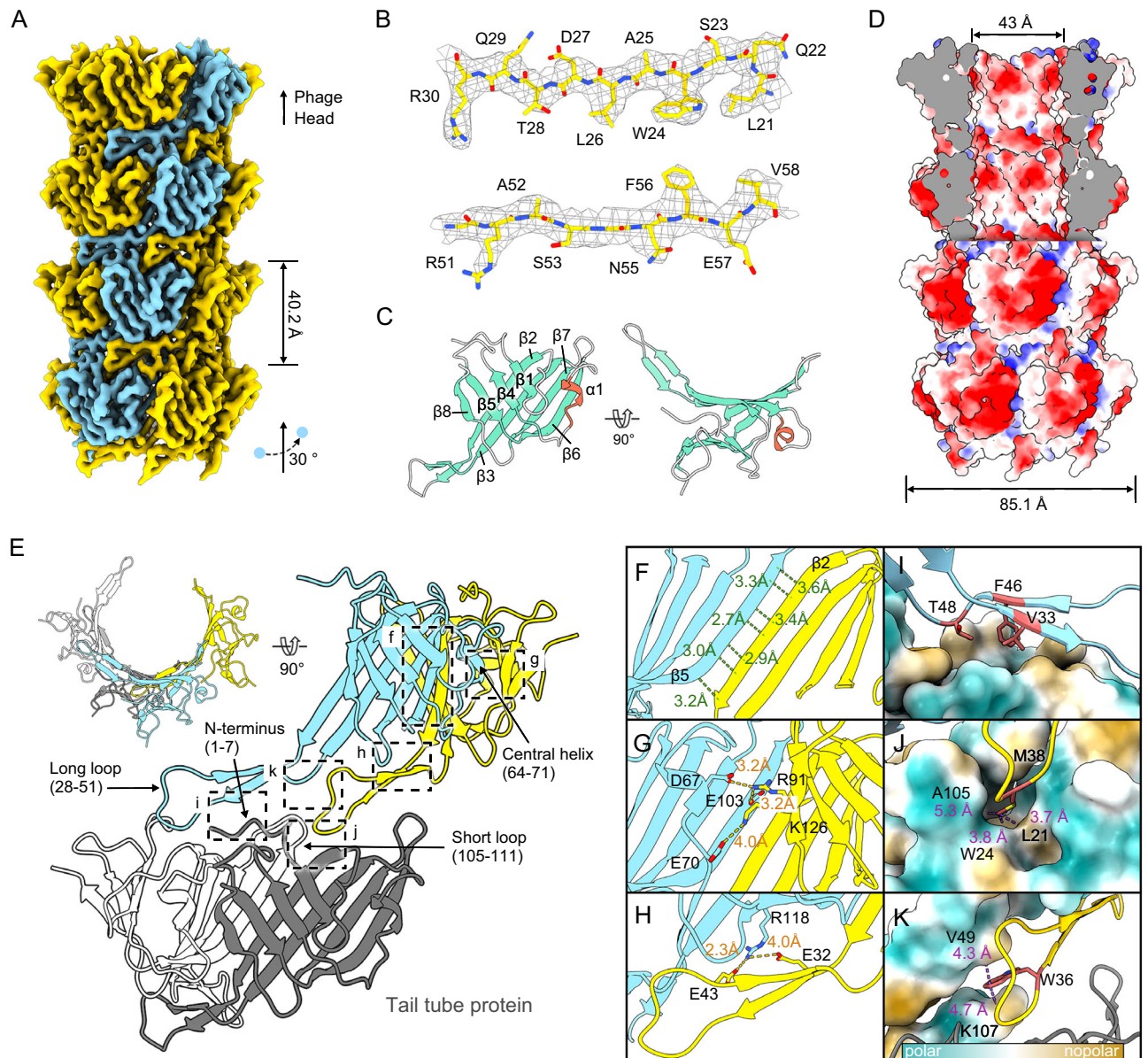

**Fig. 5 | Atomic structure of the tail tube of R4C. A** A surface representation of 3.43-Å cryo-EM map of the four-disc tail tube proteins (TTPs) is shown, highlighting one TTP helical strand in blue. Helical rise and twist are indicated. **B** Representative cryo-EM density maps (mesh) and corresponding atomic models (sticks). **C** Side (left panel) and top views (right panel) of an individual subunit of the TTP are shown in ribbon representation with the α-helices and β-strands coloured in red and turquoise, respectively. The β-strands are numbered sequentially (β1–8). **D** Side view of the tail tube (4 discs) with the front top half split and the electrostatic charge distribution on the outer and inner surfaces shown. Positive (blue), negative (red), neutral (white). The outer and inner diameters of the tail tube are labelled. **E** Relative positions and interactions of the adjacent TTPs from the same disc and from two consecutive discs. The long loop (yellow) extends underneath and

contacts the central β-barrel of the neighbouring subunit (blue) from the same disc and the underneath subunits (white and grey) of the next disc. **F–H** Details of the interactions between TTP subunits from one disc. Two adjacent subunits are stabilized by an inter-strand hydrogen bond network (**F**) and two salt bridge networks (**G, H**), as indicated by green and orange dashed lines, respectively. **I–K** Details of the interactions between TTP subunits from two consecutive discs. Interactions are mainly contributed by non-polar interactions, as shown by the purple dashed line. For clarity, the molecule showing the key non-polar residues is depicted in ribbon representation, whereas the other molecules forming the hydrophobic pocket are depicted in molecular surface representation. Hydrophobic surfaces are in yellow and charged surfaces in turquoise. Side chains of the selected residues are in stick representation.

reside next to the C-terminal domain of TMP[36] (Supplementary Fig. 12C).

We were intrigued why the R4C phage appears as a long straight tail and sought to explore its structural resilience by molecular dynamics (MD) simulation. A two-step MD was implemented on the 7-disc-length tube evicted from the R4C tail, the tail of another phage SPP1[72] with comparable length for 7-disc tube and otherwise higher flexibility was served as control. Steered MD (SMD) was firstly employed to induce bending of the tubes at four representative

directions against the tail axis (Supplementary Fig. 13A), and then conventional MD (CMD) was calculated on the bending tubes after SMD force lifted (Supplementary Fig. 13B, C). The results demonstrated that the R4C tail could nearly recover to the original straight shape after 10 ns CMD simulation at either of four directions, whereas the control SPP1 tail remains unchanged bending gesture (Supplementary Fig. 13D), which suggested the R4C tail may have excellent resilience to maintain straight shape in the native environment. On the other hands, we wondered whether the compact sixfold symmetry

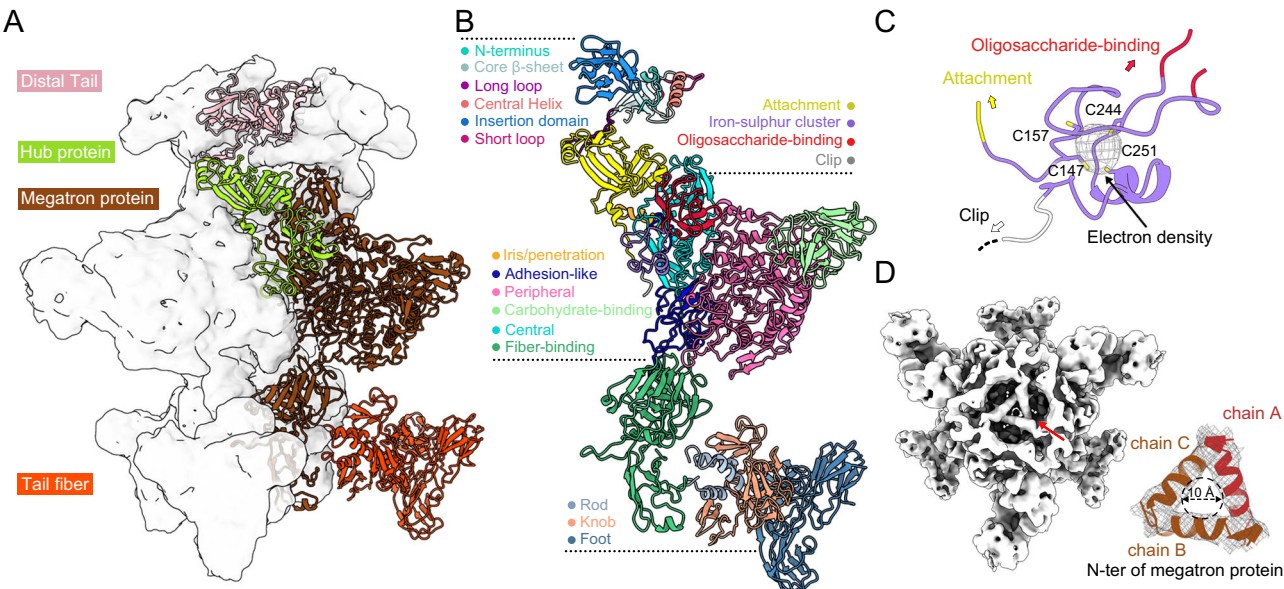

**Fig. 6 | Structure of the absorption device at the distal end of R4C. A** Gaussian-filtered C3 density map of the distal tail and baseplate is rendered in a semi-transparent white surface and the corresponding models shown and coloured according to protein components. **B** Models of each component colour-coded by domain. The undetermined clip domain is marked in grey text. **C** Detail of an iron-sulfur cluster coordinated by four Cys of the hub protein. The strong density (grey mesh) corresponding to the iron-sulfur cluster is shown. **D** Head-to-tail view of the density map of the baseplate with the iris/penetration domains of megatron proteins marked by a red arrow. The right panel shows a close-up view of the N-terminal α-helix of iris/penetration domains from three megatron proteins fitting well into the densities of the iris-like gate, with one of the helixes highlighted in red. The inner diameter (-10 Å) of the tunnel is depicted.

layer stacking nature would contribute to the structural resilience of R4C tail. The MD using the structure-based model (SMOG) instead of CMD was applied to simulate the unbending process of the SMD-induced bent R4C tube. The SMOG model is built in terms of structure-dependent properties of single molecule or multi-molecule assemblies while filtering out effects related to the intra- or inter-molecular interactions[81]. Interestingly, the R4C tail remains unchanged bending angle after SMOG-based MD, revealing that the resilience of R4C tail was attributed by molecular interactions rather than molecule stacking (Supplementary Fig. 13E).

**Structures of RcGTA-like distal tail and baseplate**

Structural reconstruction of the tail endpoint is challenging due to its low identifiability and preferred orientation in cryo-EM images. However, we were able to capture some 2D classes that show the typical features of the baseplate during helical reconstruction of the tail tube (Fig. 3C), which indicated that the tail endpoint was also picked from the filament tracer procedure. Reconstruction of these particles imposing C3 symmetry resulted in a 4.5-Å resolution map that allowed for Cα backbone model building at the end of the R4C phage tail (Supplementary Fig. 6D).

The R4C distal tail—attaching the tail tube upward and the baseplate downward—is a C6 hexamer, as revealed by C3 reconstruction. Each Dit protein can be structurally split into two domains: the core domain presents as a twisted β-sandwich structure with two β-hairpins and two helices, whereas the insertion domain protruding outward from the tube surface is organized in a barrel-like fold and decorated with two insertion loops (Fig. 6A, B). The core domain harbouring the central channel provides attachment sites for tail tube and the baseplate (Supplementary Fig. 14A). Both domains of the Dit monomer mediate inter-subunit interactions. The β1 and two insertion loops form a notch to clamp the α2 of the adjacent monomer, ensuring strength to the hexamer association (Supplementary Fig. 14B). Short loops from two adjacent monomers, and the α1 and β2-α2 loops interact with the long loop of a tail tube monomer (Supplementary Fig. 14C). The Dit long loop adopts an L-shaped β-hairpin belt to create a flexible surface for optimal binding of the baseplate, mediating symmetry mismatch between the C6 Dit hexamer and the C3 megatron-hub complex (Supplementary Fig. 14D, E).

The Dit core domain is structurally homologous to that from RcGTA and phage T5 (RcGTA: DALI analysis Z score = 8.4, sequence identity = 36%; T5: Z score = 7.3, sequence identity = 6%), suggesting their evolutionary relationship (Supplementary Fig. 9G, H)[36,82]. The Dit insertion domain from RcGTA is considered to be homologous to that of phage T5, which adopts an oligosaccharide/oligonucleotide-binding (OB-like) fold and provides host adhesion properties[36]. Yet, we observed no structural similarity for the Dit insertion domain of R4C; instead, it resembles the N-terminus of the outermost domain of the phage T7 tail fibre protein (Supplementary Fig. 15E)[83]. PSI-BLAST and Pfam analyses confirm associations between R4C Dit protein and proteins belonging to glycoside hydrolase family, which can capture saccharides[84]. Unlike T5 Dit, we speculate that R4C Dit functions as a tail adsorption apparatus, recognizing the different cell-wall receptors/components.

The C3 baseplate assembles at the distal end of the R4C tail (Fig. 3A). The overall R4C baseplate is homologous to that of RcGTA and both comprise hub (35.5% sequence identity), megatron (39.6% sequence identity), and fibre (30.4% sequence identity) proteins (Fig. 6A, B). According to established nomenclature, the R4C hub protein is divisible into four domains: the attachment (1–141), ion-binding (142–161, 244–266), oligosaccharide-binding (162–243) and clip domains (267–291) (Fig. 6B). The attachment domain presents a predominate docking interface for the distal tail (Supplementary Fig. 14D, E). The ion-binding domain, which comprises two discrete segments, uses four conserved Cys residues to coordinate an iron-sulfur cluster (Fig. 6C), which is essential for tail assembly and phage biological activity[85]. The oligosaccharide-binding domain may arise through horizontal gene transfer, and the saccharide-binding property—presumably host specific—enhances cell adhesion.

The megatron protein is divided into six domains: iris/penetration (1–45), adhesion-like (46–211), peripheral (212–481, 630-903), carbohydrate-binding (482–629), central (904–1130) and fibre-binding

domains (1131–1447) (Fig. 6B). The iris/penetration domain is highly flexible except for its N-terminal α1 helices, which correspond to a noticeable density inner the baseplate channel (Fig. 6D). The three α1 helices from the three megatron proteins form an iris-like gate in a closed state (inner diameter, ~10 Å) that resembles that of RcGTA[36]; this structure aids in preventing the eruption of genomic DNA. The iris/penetration domain enables transmembrane helix formation (Supplementary Fig. 15A, B), which may function to penetrate the outer membrane and create a pore for cargo transport. We speculate that both the membrane-spanning helices from the megatron and the TMPs create the transmembrane channel for DNA translocation across the different layers of the bacterial membrane. The central domain—sharing the same topology as the core domain of the distal tail—mainly participates in distal tail binding (Supplementary Fig. 14D, E). Below the central domain is the adhesin-like domain, where the tail tunnel terminates. The adhesion-like domain is partially highly flexible and includes a disordered loop (residues 95–113) (Supplementary Fig. 15A, C). Evolutionarily, the peripheral domain is predicted to fold into a mannanase-like structure (Supplementary Fig. 15F, G). Furthermore, compared with RcGTA, an extra domain is found extending beyond the peripheral domain (Supplementary Fig. 15A). We designate this domain as a carbohydrate-binding domain, as it shares a similar topology with the carbohydrate-binding module of *Clostridium thermocellum* (CtCBM11), which binds various poly- and oligo-saccharides (Supplementary Fig. 15H, I)[86]. Lastly, the fibre-binding domain at the bottom consists of three separate regions: two provide an interface for the binding of fibre proteins, while the third may be committed to maintain the base of baseplate in a meta-stable state (Fig. 6B).

Each of the three tripod-like trimeric fibre complexes is formed by a combination of three tail fibre monomers, in which the head (1–35), body (36–117), and foot (118–230) regions are referred to as the rod, knob, and foot domains, respectively (Fig. 6A, B). The rod domain is a coiled-coil domain with an N-terminal extension binding to the megatron fibre-binding domain (Fig. 6B). The knob domain connects the rod to the C-terminal foot, which is a β-sheet-rich domain with a lectin-like fold (an 8-stranded anti-parallel beta-barrel), able to bind carbohydrates (Fig. 6B)[87].

Many marine bacteria, including roseobacters, produce capsular polysaccharides to enhance their competition and adaptation to marine environments[88,89]; this simultaneously provides an abundance of carbohydrates for phage recognition. Accordingly, the R4C phage equips various carbohydrate-binding accessories, including the Dit insertion domain, the megatron carbohydrate-binding and peripheral domains, the hub oligosaccharide-binding domain and the fibre foot domain to attach to the bacteria capsule. These specific structural components are essential for phage infection via facilitating bacterial host anchoring.

## Ecological perspective on the relationships between phage and GTA

The GTA is thought to originate from (pro)phage based on genomic analysis, particularly genes encoding tail-related proteins[90,91]. We noted high structural similarity between the tail-related proteins from R4C and RcGTA: this provides further evidence for the evolutionary correlation between the GTA and bacteriophages and allows us to propose an ecological perspective for their relationship. GTA expression and secretion are performed by bacteria to maintain the host population[90,91]. GTA-encoding genes in the host may be obtained from phages, as the integrases (integrating enzyme) and repressors enable phages to integrate their genomes into the host genomes thereby allowing the host to acquire profitable genes from phages for GTA production and evolution. Nevertheless, phylogenetic analysis shows that GTA genes in *D. shibae* DFL12 and their specific phage genes, including those from R4C, are not closely related (Supplementary Fig. 16), suggesting a separate evolution in bacteria and phages.

Reciprocally, under certain circumstances, phages can obtain GTA-encoding genes from the host. Most tail-related proteins from the R4C phage demonstrate structural, and possible functional, conservation with the GTA, from which phages may benefit. More specifically, the good growth conditions of the host bacteria[92,93], which are critical for high GTA expression and secretion, are also suitable for viral infection and production. On the other hand, GTA-mediated horizontal gene transfer occurs much more widely (across phylum) than its host range[94], implying that GTA can enter a wider range of bacteria. With these considerations in mind, roseophages, such as R4C, which have similar attachment, adsorption, entry, and DNA delivery mechanisms, may have the potential to win the virus-host evolutionary arms race and show a higher ecological advantage over other phages.

## The proposed DNA delivery mechanism of the R4C phage

Based on the in-situ structure of the intact R4C virion, along with our comparative analysis of viral proteins and their homologous structures, we propose a DNA delivery mechanism of the R4C phage (Fig. 7). We surmise that the R4C phage first adheres to host bacteria by establishing interactions with the specific bacterial receptor (e.g., capsular polysaccharides) or its co-factors using the extra domains of the distal tail proteins and/or the megatron proteins (Supplementary Fig. 15A). The three trimeric fibre proteins, function as anchors and orient the phage, then drive the baseplate towards the outer membrane to attach tightly to the bacteria host. Cascading conformational rearrangements within the baseplate, and even the whole tail, then occur to further trigger opening of the tail tunnel from its closed state. Hereafter, the hydrophobic N-termini of the megatron proteins, which form the latch (Fig. 6D), are exposed and then insert themselves into the bacteria outer membrane to form a transmembrane pore. Release of the peptidase then leads to degradation of the peptidoglycan layer between the bacteria outer and inner membranes. The TMPs exit the tail and enter the periplasmic space, presumably spanning the bacterial cell envelope and forming a conduit through the inner membrane using their transmembrane segments (Supplementary Fig. 12); this transport may require the help of glucose transporter proteins[78]. TMP egress may in turn trigger signal transduction from the terminator to the portal, which appears as a series of sequential structural changes (e.g., the portal tunnel loop) that finally initiate genome translocation. The pressure stored in the phage head propels the genomic DNA into the channel of the tail tube, but this may be insufficient to move the complete genome against the internal osmotic pressure of the bacterial cell[95,96]. The negative charged tracking on the interior surface of the tail tube (Supplementary Fig. 11), complemented by its straight tubular architecture (Fig. 5), will assist passage of the DNA through the long tail. Finally, the genome is ejected into the host bacteria crossing the channel formed by the TMPs.

In summary, genome release of R4C as well as other tailed phages is accomplished via a complex collaborative process mediated by various structural proteins. Additional details to complete the procedure need to be further uncovered.

## Methods

### Preparation and purification of the R4C phage

The host bacterium *D. shibae* DFL12 was grown at 28 °C in RO medium (1 g/L yeast extract, 1 g/L peptone, and 1 g/L sodium acetate at pH 7.5), with shaking at 160 rpm. The vB_DshS-R4C stock in SM buffer (50 mM Tris-HCl pH 7.5, 100 mM NaCl, and 8 mM MgSO₄) was added to DFL12 when the OD600 of host was equal to 0.2–0.3. After 24-h infection, the lysate was centrifuged at 8000 rpm for 10 min and filtered through 0.2-μm polycarbonate membrane (Merck Millipore), and the aqueous phase was precipitated with 10% (w/v) dissolved polyethylene glycol (PEG) 8000 and NaCl (1 M) at 4 °C for 30 h. The mixture was then centrifuged at $10,000 \times g$ for 50 min at 4 °C, and the pellet was gently resuspended in 5 mL of SM buffer and purified with a CsCl₂ density

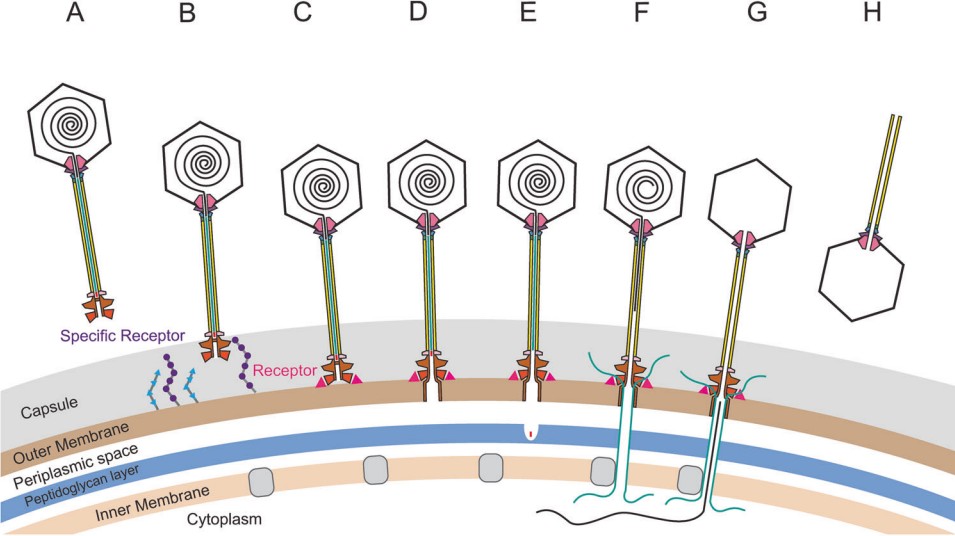

**Fig. 7 | Proposed mechanism of the R4C genome delivery into *D. shibae* DFL12ᵀ.**
**A** The mature R4C virion. **B** The R4C virion recognizes and attaches to the host receptor or co-factor via its distal tail protein or baseplate. **C** The tail fibre proteins enable R4C to further bind to the host surface using its tail axis. **D** Penetration of the outer membrane by the megatron protein iris/penetration domain. **E** Degradation of the host peptidoglycan layer by cell-wall peptidase. **F** The tape measure proteins (TMP) are released and form a channel throughout the host membrane. **G** Ejection of the genome into the host cytoplasm through the TMP channel. **H** The empty R4C particle.

gradient (1.3 mg/mL, 1.5 mg/mL, 1.7 mg/mL; 34100 × *g*, 4 °C, 24 h). Purified phage particles were collected and dialyzed twice with SM buffer overnight in the dark at 4 °C.

### Cryo-EM sample preparation and data collection
Aliquots of 3 µL of purified R4C phage were loaded onto glow-discharged (60 s at 20 mA) holey carbon Quantifoil grids (R1.2/1.3, 200 mesh, Quantifoil Micro Tools) using a Vitrobot Mark IV (Thermo Fisher Scientific) at 100% humidity and 4 °C. Data were acquired on a FEI Tecnai F30 transmission electron microscope (Thermo Fisher Scientific) operated at 300 kV and equipped with a Falcon 3 direct electron detector. Images were recorded in the 39-frame movie mode at a nominal ×93,000 magnification with a pixel size of 1.12 Å. The total electron dose was set to 30 e⁻ Å⁻² and the exposure time was 1 s. Data were automatically collected with Thermo Fisher EPU software.

### Image processing and 3D reconstruction for the capsid, tail and baseplate
Drift and beam-induced motion correction was performed with MotionCor2[97] to produce a micrograph from each movie. Contrast transfer function (CTF) fitting and phase-shift estimation were conducted with Gctf[98]. Micrographs with astigmatism, obvious drift, or contamination were discarded. The following reconstruction procedures were performed by using Cryosparc V3[99]. In brief, particles of R4C head were automatically picked using the "Blob picker" and then "Template picker", while the "Filament tracer" was used for the cylindrical R4C tail to facilitate density particle picking along the tail axis. Several rounds of reference-free 2D classifications on two datasets yielded two distinct types of classes, of which one represented the tail tube and the other, the distal tail of the R4C tail. Selected good particles of the capsid and distal tail were both subjected to ab-initio reconstruction and homogeneous refinement imposing I2 and C3 symmetry, respectively. Particles of the tail tube were subjected to several iterations of helical reconstruction and symmetry search process to precisely determined the symmetry, rise and twist of the helical tail, eventually obtaining the density map. The resolution of density maps for the capsid and tail tube were determined by the gold-standard Fourier shell correlation curve, with a cut-off of 0.143[100]. Local map resolution was estimated with ResMap[101].

### Image processing and 3D reconstruction for the R4C tail neck
The images preprocessed in Cryosparc were imported to relion 3.1 for further reconstruction of the R4C neck. Briefly, hundreds of particles of R4C head were picked manually and subjected to 2D classification (bin2), the best-defined 2D class averages were used to autopick particles in all micrographs. 2D classification and 3D refinement (bin2) were performed to obtain the I3 reconstruction of the capsid. 14,776 particles were then extracted (bin1) and subjected to the symmetry expansion in Relion using I3 symmetry. Sub-particles from the fivefold vertexes were re-extracted; those representing the C12 portal vertex was classified out using 3D classification imposing C5 symmetry. To determine the neck proteins, further symmetry expanded (C5) and 3D (C12) classifications were performed, yielding five classes of the portal vertex with different global orientations differing by 72°; all these five classes presented a typical feature of the dodecameric portal and adaptor. Based on particles from one of the five classes, localized sub-particle reconstructions using different symmetries and masks were carried out to obtain the density maps of the C1 portal vertex, the C5 portal vertex, the C6 portal vertex and the C12 portal vertex, respectively.

### Atomic model building, refinement, and 3D visualization
The initial model of the R4C MCP was generated from homology modelling based on the atomic model of the MCP of HK97 (PDB ID: 1OHG) by Accelrys Discovery Studio software (available from: URL: https://www.3dsbiovia.com). For tail components, their sequences were subjected to the protein structure prediction server trRosetta (available from URL: https://yanglab.nankai.edu.cn/trRosetta/)[48], with the predicted models serving as the initial model for further refinement. We initially fitted the templates into the corresponding final cryo-EM maps using Chimera[102], in detail, the capsid proteins were fitted into the I2 head map, the portal and adaptor proteins into the C12 neck map, the stopper and terminator proteins into the C6 neck map, the tail tube proteins into the C6 tail map, the baseplate complex, distal tail and tail fibre proteins into the C3 base map, respectively. Every models were corrected and adjusted manually by real-space refinement in Coot[103]. As the R4C neck and base exhibited weak densities, we built the backbone models only for portal, adaptor, stopper, terminator, distal tail, megatron, hub, tail fibre proteins. For the major

capsid and tail tube proteins, the resulting models were further refined with phenix_real_space_refine in PHENIX[104]. These operations were executed iteratively until the problematic regions, Ramachandran outliers, and poor rotamers were either eliminated or moved to favoured regions. The totality of these seven asymmetric units of capsid were subjected to further real-space refinement to optimize the clashes. The final atomic models were validated with Molprobity[105,106]. Sequence alignment was performed with Clustal Omega on the EBI server (https://www.ebi.ac.uk/Tools/msa/clustalo/). All figures were generated with Chimera or ChimeraX[107,108].

### Molecular dynamics simulation

Steered molecular dynamics (SMD) simulations were first employed, serving as a model-building tool to artificially generate bent structure of the R4C and SPP1 tail tubes. The final structures extracted from SMD simulations were subjected to conventional MD (CMD), with the force removing to estimate the elasticity of the tail by tracking the bending angle changes of the corresponding tail tube. To achieve this, atomic models composed of seven discs of the R4C and SPP1 (PDB ID: 6YQ5) tail tube proteins were used. The prepared models were submitted to Autopsf in VMD[109] to generate topology information. The structure was embedded in the explicit solvent (water) box encompassing 12 Å from the protein boundary using the TIP3P (transferable inter-molecular potential with three interaction sites) water potential model[110]. The system was neutralized by adding counter chloride ions to achieve zero charge followed by additional sodium and chloride ions to a final physiological concentration of 0.15 M. We applied CHARMM36 forcefield in our simulation, and periodic boundary conditions were employed to avoid an edge effect.

The two prepared systems were submitted to SMD simulation with the force applied to CA atoms from the middle five layers of the seven discs tail tube. The applied forces decrease from the middle layer to both flanking layers to induce a continuous bend. The force applied to the middle layers was set at 2.08, 1.38 kcal/mol/Å for the 3rd/5th layers, and 0.69 kcal/mol/Å for the 2nd/6th layers. The tail tube was induced to bend at four representative directions (0°, 15°, 30°, 45°) between two subunits (with 60° interval) of the sixfold symmetry layer, the directions were against the tail axis (Supplementary Fig. 13A). Of these, the direction at 0° is pointing from centre of mass (COM) of one chain to that of the opposing chain. Each SMD simulation was performed with 150 picosecond (ps) duration, and then additional 10 nanosecond (ns) CMD simulations were employed. Each CMD simulation was repeated five times. The bending angle reflecting the 7-disc tube shape was defined by the three points, which is corresponding to COMs of the top, middle and bottom layers of the tube. The potential variation of bending angles will be monitored throughout the MD.

MD simulations were performed using the NAMD version 2.13 MD package[111]. The integration timestep of the simulation was set to 1 femtosecond (fs) and the position coordinates (DCD file) were saved every 0.2 ps for SMD, 4 ps for CMD, for further analysis. Long-range periodic electrostatic interactions were evaluated using the smooth Particle-Mesh Ewald (PME)[112] method, with a real cut-off radius of 10 Å. Lengths of all chemical bonds involving hydrogen bonds were constrained by the SHAKE algorithm[113]. CMD simulations were performed at 310 K and constant temperature was controlled by Langevin dynamics[114] under a pressure of 1 atm[115] maintained using the Nose-Hoover thermostat. The constant temperature control was switched off in SMD to decrease the freedom of the atoms. Before each production, the system was energy-minimized by 2000 conjugate gradients steps to reduce steric conflicts between water molecules and the protein. The dynamic results were analyzed using the VMD programme.

The simulation based on all-atom SMOG−a native-centric model retaining the full atomic structural details−was performed to explore the weight of molecular structure contributed to the resilience of phage R4C tail. The tube structure from the SMD simulation (0°) was subjected to the SMOG-model web server[116] to generate the forcefield file. MD simulations were performed for $1.0 \times 10^8$ timesteps of duration 0.0005 reduced units (50 ns in total) using the Gromacs (v4.6.7)[117]. Langevin Dynamics protocols were used to ensure a constant temperature of 0.4 (reduced units). Each SMOG-based MD simulation was repeated five times.

### Reporting summary

Further information on research design is available in the Nature Portfolio Reporting Summary linked to this article.

## Data availability

The data that support this study are available from the corresponding authors upon request. The cryo-EM density maps have been deposited in the Electron Microscopy Data Bank (EMDB) with the accession codes of EMD-34247 (capsid), EMD-34253 (C1 portal vertex), EMD-34254 (C5 portal vertex), EMD-34250 (C12 portal vertex), EMD-34252 (C6 stopper-terminator), EMD-34248 (tail tube), EMD-34249 (distal tail and baseplate), and the corresponding atomic coordinates have been deposited in the Protein Data Bank (PDB) with the accession codes of 8GTA (capsid), 8GTD (C12 portal-adaptor), 8GTF (C6 stopper-termi-nator), 8GTB (tail tube), 8GTC (distal tail and baseplate). Atomic coordinates of previously determined structures are available in the PDB under the following accession codes: 1OHG, 5WK1, 6TSU, 5LII, 5VF3, 6TB9, 6J3Q, 7VIK, 6XGQ, 6I9E, 6QVK, 5UU5, 5L35, 3JA7, 6QX5, 6IBG, 6QJT, 6TE8, 2KX4, 2KCA, 3F3B, 6TE9, 2LFP, 3FZ2, 6YQ5, 5W5F, 6TSV, 4JMQ, 6TEH, 2POH, 7BOZ, 7DVJ, 2LRP. The MD simulation data generated in this study have been deposited in the Zenodo OpenAIRE database under accession code 7947658. Source data are provided with this paper.

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

## Acknowledgements

This study was supported by the National Natural Science Foundation of China (grants nos. 42188102 and 91951209 to R.Z., no. 32170942 to Q.Z. and no. 81991491 to N.X.).

## Author contributions

Q.Z., S.L., R.Z. and N.X. conceived the project. Y.H., H.S. and S.W. designed the experiments. Y.H., H.S., S.W., L.C. and J.X. performed the experiments. L.L., Z.C. and Y.J. collected the cryo-EM data. Y.H., H.S., Q.Z. and S.L. determined the structures and made figures. Y.Q., Z.K., T.L., H.Y., J.Z. and Y.G. participated in the discussion and interpretation of the results. S.L., Q.Z., R.Z., Y.H., H.S. and S.W. wrote the manuscript. All authors contributed to the data analysis and manuscript preparation.

## Competing interests

The authors declare no competing interests.
