## [Peer Review File · Nature Communications]

Structure and proposed DNA delivery mechanism of a marine roseophageReviewers' Comments:

Reviewer #1:

Remarks to the Author:

This study describes cryo-EM reconstructions of a complete marine roseophage. This structure is complemented by molecular dynamics simulations of the tail. In the simulations a force is applied to induce a bent structure for the R4C and SPP1 tails. The authors induced an approximately 10 degree bend in each, and then released the applied force to probe the responsiveness of the tails. They find that the R4C tail returns to the straight orientation rapidly, suggesting it is more stiff than the SPP1 tail. As written, the calculations provide anecdotal support, though some additional simulations could provide a more conclusive insight.

Major points:

1) It is not clear if the results are reproducible. While the R4C tail returned to a straight orientation faster in a single run, it is not clear this is always the case. The simulations should be repeated at least 5-10 times, in order to establish some statistical significance of the results. The initial claim may end up being correct, though additional simulations may very well show that the first event was an anomaly.

2) Related to the simulations, another question is whether the direction of bending is important. That is, there is the claim that one tail is more stiff. However, at short length scales, stiffness may be anisotropic. Accordingly, the higher apparent stiffness of R4C may be due to the choice of bending direction, where one direction in R4C is more stiff than a different direction in SPP1. Of course, since the proteins form a superhelical tail structure, then at very long length scales the stiffness would be isotropic. However, here, the bending is isolated over a single layer of proteins within the tail (more of a "kink" than a continuous bend). Due to this very short length scale over which bending is probed, it is important to fully understand the potential anisotropies.

3) The bending angle is not clearly defined. I searched the SI and main text, but did not find a definition (only a graphical depiction). Perhaps it is buried in the text, or missing. Please provide an exact definition of the angle, so that the calculations could be reproduced.

4) It would be interesting to see if the results are due to the details of the energetics, or simply the structure of the tail. There are many simplified models, such as all-atom structure-based SMOG models (Go-like models) that can be used to probe how structure contributes to mechanical properties. If these simpler models also suggest that R4C is more stiff, it would indicate that the arrangement of proteins is the source of stiffness. If SMOG models do not indicate differential stiffness, and the explicit-solvent simulations do, then this would suggest the specific interactions (e.g. electrostatic or solvation interactions) may be the origin of differential stiffness. For a system of this size, these types of simulations could probably be performed in a day using a desktop, so they would be a computationally inexpensive way to elaborate on the origins of stiffness.

Minor points

The term "Steered MD" is not the most suitable way to describe the simulations. While an applied force is introduced to build models of the bent structures, the focus of the analysis is not the bending, but the unbending process. Accordingly, the manuscript doesn't actually describe the SMD simulations, except to note that the force induced a bend. It would be more descriptive to state that an applied force was used to induce a bend, and then unrestrained simulations were used to probe stiffness. As written, the reader will expect that SMD was used to learn something about bending, but SMD was only a model-building tool, in this case.

Reviewer #2:

Remarks to the Author:

The manuscript "Structure and proposed DNA delivery mechanism of a marine roseophage" by Huang et al. reported the overall structure of marine rose phage R4C and the in situ structure of the long tail by cryo-EM single-particle analysis. There, the specialized structure of the long tail, consisting of portal, adapter, stopper, terminator, tail tube, distal tail, baseplate and tail fiber, proposed a mechanism of infection. The atomic structures of the individual components explain the symmetry discrepancies between these, and the negative charge distribution within the tail tube drives the movement of viral DNA within the long tube, in addition to the repulsive forces of the packed DNA within the capsid. Structural comparison of each component of similar phages and phage-like gene transfer agents confirmed the infection mechanism and evolution to the bacterial host. The manuscript written by comprehensive English is easily accessible not only to professional microbiologists, but also to a wide range of readers. For these reasons, it is suitable as research article for this journal. However, the authors need to clarify a few things I commented below before accepting.

Major comments:

The reviewer is concerned with how precisely the atomic structures of the individual tail components have been determined, as validation information is not presented for all components in Supplementary Table 1.

Minor comments:

1) P. 4 Line 88: "the previously identified ratchet mechanism that assists in DNA transmission" It needs a reference.

2) References 35 and 39 are the same.

3) Fig. 4 Legend, P. 30 Line 883: and he adaptor -> and the adaptor

4) Fig. 4 Legend, P. 30 Line 893: Inter-layer -> inter-layer

5) Fig. 6 Legend, P. 34 Line 927: red arrow -> black arrow

6) Fig. S6: "A. Representative cryo- EM micrograph used for tail reconstruction." The micrograph is missing.

Response to Reviewer Comments on the manuscript NCOMMS-22-37124:

We thank the reviewers for their constructive comments to improve our manuscript. We have revised the manuscript with additional MD simulations and analysis. To facilitate the navigation of this document, we have copied the reviewers' comments verbatim in blue and typed out our responses in black.

Reviewer #1

Comments to the Author

Reviewer: This study describes cryo-EM reconstructions of a complete marine roseophage. This structure is complemented by molecular dynamics simulations of the tail. In the simulations a force is applied to induce a bent structure for the R4C and SPP1 tails. The authors induced an approximately 10 degree bend in each, and then released the applied force to probe the responsiveness of the tails. They find that the R4C tail returns to the straight orientation rapidly, suggesting it is more stiff than the SPP1 tail. As written, the calculations provide anecdotal support, though some additional simulations could provide a more conclusive insight.

Response: We thank the reviewer for raising comments regarding to MD simulations. As suggested, we performed comprehensive MD simulation on recapitulating the unbending process of R4C tail.

Major comment 1: It is not clear if the results are reproducible. While the R4C tail returned to a straight orientation faster in a single run, it is not clear this is always the case. The simulations should be repeated at least 5-10 times, in order to establish some statistical significance of the results. The initial claim may end up being correct, though additional simulations may very well show that the first event was an anomaly.

Response: We thank the reviewer for spot-on points about MD calculation. As suggested, we runned the simulation of unbending process with 5 times repeat for every condition, which show similar recovering trend over simulation time as plotted by mean bending angle

and standard deviation. Please refer to the updated Supplementary Figure 13D.

Major comment 2: Related to the simulations, another question is whether the direction of bending is important. That is, there is the claim that one tail is more stiff. However, at short length scales, stiffness may be anisotropic. Accordingly, the higher apparent stiffness of R4C may be due to the choice of bending direction, where one direction in R4C is more stiff than a different direction in SPP1. Of course, since the proteins form a superhelical tail structure, then at very long length scales the stiffness would be isotropic. However, here, the bending is isolated over a single layer of proteins within the tail (more of a "kink" than a continuous bend). Due to this very short length scale over which bending is probed, it is important to fully understand the potential anisotropies.

Response: We thank the reviewer for raising the great and constructive concern. To explore whether the bending stiffness against various directions around the round tail is anisotropic or not, we performed the unbending process simulations at four representative bending directions (0° , 15° , 30° , 45°) between two subunits (with 60° interval) of the six-fold symmetry layer. We agree with the reviewer that the bending in short length scale should be a continuous bend rather than a "kink", so we applied a gradient force decreasing from the middle layer to both its flanking layers during steered-MD simulations at the above-mentioned four directions (please refer to the updated Supplementary Figure 13A). After the SMDs, the systems were subjected to conventional MD with 5 times repeat. The overall profiles of both cases showed the R4C tail could nearly recover to the original straight shape after 10 ns simulation, whereas the control SPP1 tail remains unchanged bending angles (please refer to the updated Supplementary Figure 13D).

Major comment 3: The bending angle is not clearly defined. I searched the SI and main text, but did not find a definition (only a graphical depiction). Perhaps it is buried in the text, or missing. Please provide an exact definition of the angle, so that the calculations could be reproduced.

Response: Sorry for the missing information. We defined the bending angle in the Methods section. "The bending angle reflecting the 7-disc tube shape was defined by the three

points, which is corresponding to COMs of the top, middle and bottom layers of the tube. " (Page 20, Lines 579-581).

Major comment 4: It would be interesting to see if the results are due to the details of the energetics, or simply the structure of the tail. There are many simplified models, such as all-atom structure-based SMOG models (Go-like models) that can be used to probe how structure contributes to mechanical properties. If these simpler models also suggest that R4C is more stiff, it would indicate that the arrangement of proteins is the source of stiffness. If SMOG models do not indicate differential stiffness, and the explicit-solvent simulations do, then this would suggest the specific interactions (e.g. electrostatic or solvation interactions) may be the origin of differential stiffness. For a system of this size, these types of simulations could probably be performed in a day using a desktop, so they would be a computationally inexpensive way to elaborate on the origins of stiffness.

Response: We thank the reviewer for suggesting a way to elaborate on the origins of stiffness of R4C tail. As suggested, we generated the all-atom SMOG model for the bent R4C tail induced by the above-mentioned SMD. Then we simulated the unbending process using Gromacs (<https://www.gromacs.org>). The results suggest the origins of R4C tail stiffness is contributed by energetic interaction instead of the structure of the tail per se. We have added the results (new Supplementary Figure 13E) and rephrased our previous presumption, which now reads: "On the other hands, we wondered whether the compact 6-fold symmetry layer stacking nature would contribute to the structural resilience of R4C tail. The MD using the structure-based model (SMOG) instead of CMD was applied to simulate the unbending process of the SMD-induced bent R4C tube. The SMOG model is built in terms of structure-dependent properties of single molecule or multi-molecule assemblies while filtering out effects related to the intra- or inter-molecular interactions. Interestingly, the R4C tail remains unchanged bending angle after SMOG-based MD, revealing that the resilience of R4C tail was attributed by molecular interactions rather than molecule stacking (Supplementary Fig. 13E)." (Page 12, Lines 351-358)

Minor comment 1: The term "Steered MD" is not the most suitable way to describe the

simulations. While an applied force is introduced to build models of the bent structures, the focus of the analysis is not the bending, but the unbending process. Accordingly, the manuscript doesn't actually describe the SMD simulations, except to note that the force induced a bend. It would be more descriptive to state that an applied force was used to induce a bend, and then unrestrained simulations were used to probe stiffness. As written, the reader will expect that SMD was used to learn something about bending, but SMD was only a model-building tool, in this case.

Response: We agree with the reviewer that the steered MD is served as a model-building tool to induce a bend for unbending process simulation in our study. As suggested, we now rephrase the sentences for the SMD simulations, which reads: "Steered MD (SMD) was firstly employed to induce bending of the tubes at four representative directions against the tail axis (Supplementary Fig. 13A), and then conventional MD (CMD) was calculated on the bending tubes after SMD force lifted (Supplementary Fig. 13B and 13C)." (Page 12, Lines 345-348), and "Steered molecular dynamics (SMD) simulations were first employed, serving as a model-building tool to artificially generate bent structure of the R4C and SPP1 tail tubes." (Page 19, Lines 560-561)

Reviewer #2

Comments to the Author

Reviewer: The manuscript "Structure and proposed DNA delivery mechanism of a marine roseophage" by Huang et al. reported the overall structure of marine rose phage R4C and the in situ structure of the long tail by cryo-EM single-particle analysis. There, the specialized structure of the long tail, consisting of portal, adapter, stopper, terminator, tail tube, distal tail, baseplate and tail fiber, proposed a mechanism of infection. The atomic structures of the individual components explain the symmetry discrepancies between these, and the negative charge distribution within the tail tube drives the movement of viral DNA within the long tube, in addition to the repulsive forces of the packed DNA within the capsid. Structural comparison of each component of similar phages and phage-like gene transfer agents confirmed the infection mechanism and evolution to the bacterial host. The manuscript written by comprehensive English is easily accessible not only to professional

microbiologists, but also to a wide range of readers. For these reasons, it is suitable as research article for this journal. However, the authors need to clarify a few things I commented below before accepting.

Response: We thank the reviewer for the encouraging comments.

Major Comment 1: The reviewer is concerned with how precisely the atomic structures of the individual tail components have been determined, as validation information is not presented for all components in Supplementary Table 1.

Response: Thank you for the critical point. The reason for the validation information of some models not presented in the Supplementary Table1 is due to their relatively lower resolution, we could only trace their C α backbone models without side chains. For clarity, we have explained the validation unavailability for these models in the footnotes, now reads "NA - not available due to backbone model traced in lower-resolution map" in the updated Supplementary Table 1. Additionally, we provide detailed information for all the structural proteins including coding gene, domain assignment, cryo-EM map quality and built model form in a new Supplementary Table 2.

Minor Comment 1: P. 4 Line 88: "the previously identified ratchet mechanism that assists in DNA transmission" It needs a reference.

Response: Cited as *Reference 35*.

Minor Comment 2: References 35 and 39 are the same.

Response: The reference 39 has been removed.

Minor Comment 3: Fig. 4 Legend, P. 30 Line 883: and he adaptor -> and the adaptor

Response: Corrected.

Minor Comment 4: Fig. 4 Legend, P. 30 Line 893: Inter-layer -> inter-layer

Response: Corrected.

Minor Comment 5: Fig. 6 Legend, P. 34 Line 927: red arrow -> black arrow

Response: Corrected.

Minor Comment 6: Fig. S6: "A. Representative cryo-EM micrograph used for tail reconstruction." The micrograph is missing.

Response: Thank you for pointing this out. The representative cryo-EM micrograph has been presented in the Supplementary Figure 1A. Then the image will not be presented in the Supplementary Figure 6 to avoid repeat, the corresponding legend has also been removed.

Reviewers' Comments:

Reviewer #1:

Remarks to the Author:

My apologies for the delayed review. The authors have addressed all concerns adequately, and I would recommend publication.

Reviewer #2:

Remarks to the Author:

The authors have addressed all of concerns from the reviews. I recommend that this study is published. Before that, it should be corrected the following minor issues.

1) P.17 Line 484: rpm/min -> rpm

2) Fig. 4C: Stronge density -> Strong density

3) Fig. 6 Legend: "Undetermined structures are illustrated by grey circles." I cannot find the grey circle.

Response to Reviewer Comments on the manuscript NCOMMS-22-37124:

We thank the reviewers for their constructive comments to improve our manuscript. We have revised the manuscript to address their concerns. To facilitate the navigation of this document, we have copied the reviewers' comments verbatim in blue and typed out our responses in black.

Reviewer #1

My apologies for the delayed review. The authors have addressed all concerns adequately, and I would recommend publication.

Response: We sincerely appreciate the reviewer for his/her valuable comments on our study which substantially improve our manuscript.

Reviewer #2

The authors have addressed all of concerns from the reviews. I recommend that this study is published. Before that, it should be corrected the following minor issues.

1) P.17 Line 484: rpm/min -> rpm

Response: Done.

2) Fig. 4C: Stronge density -> Strong density

Response: Done.

3) Fig. 6 Legend: "Undetermined structures are illustrated by grey circles." I cannot find the grey circle.

Response: Sorry for our negligence. The description has been revised as "the undetermined clip domain is marked in gray text".